# Effects of plyometric training on technical skill performance among athletes: A systematic review and meta-analysis

**Nuannuan Deng**[1]*, **Kim Geok Soh**[1]*, **Borhannudin Abdullah**[1], **Dandan Huang**[2], **Wensheng Xiao**[3], **Huange Liu**[1]

**1** Department of Sports Studies, Faculty of Educational Studies, Universiti Putra Malaysia, Selangor, Malaysia, **2** College of Physical Education, Chongqing University, Chongqing, China, **3** Department of Sports Sciences, Huzhou University, Huzhou, China

* Dengnuannuan117@gmail.com (ND); kims@upm.edu.my (KGS)

## Abstract

### Background

The literature has proven that plyometric training (PT) improves various physical performance outcomes in sports. Even though PT is one of the most often employed strength training methods, a thorough analysis of PT and how it affects technical skill performance in sports needs to be improved.

### Methods

This study aimed to compile and synthesize the existing studies on the effects of PT on healthy athletes' technical skill performance. A comprehensive search of SCOPUS, PubMed, Web of Science Core Collection, and SPORTDiscus databases was performed on 3rd May 2023. PICOS was employed to establish the inclusion criteria: 1) healthy athletes; 2) a PT program; 3) compared a plyometric intervention to an active control group; 4) tested at least one measure of athletes' technical skill performance; and 5) randomized control designs. The methodological quality of each individual study was evaluated using the PEDro scale. The random-effects model was used to compute the meta-analyses. Subgroup analyses were performed (participant age, gender, PT length, session duration, frequency, and number of sessions). Certainty or confidence in the body of evidence was assessed using the Grading of Recommendations Assessment, Development, and Evaluation (GRADE).

### Results

Thirty-two moderate-high-quality studies involving 1078 athletes aged 10–40 years met the inclusion criteria. The PT intervention lasted for 4 to 16 weeks, with one to three exercise sessions per week. Small-to-moderate effect sizes were found for performance of throwing velocity (i.e., handball, baseball, water polo) (ES = 0.78; $p < 0.001$), kicking velocity and distance (i.e., soccer) (ES = 0.37–0.44; all $p < 0.005$), and speed dribbling (i.e., handball, basketball, soccer) (ES = 0.85; $p = 0.014$), while no significant effects on stride rate (i.e.,

**Data Availability Statement:** All relevant data are within the manuscript and its Supporting Information files.

**Funding:** The authors received no specific funding for this work.

**Competing interests:** The authors have declared that no competing interests exist.

running) were noted (ES = 0.32; $p$ = 0.137). Sub-analyses of moderator factors included 16 data sets. Only training length significantly modulated PT effects on throwing velocity (> 7 weeks, ES = 1.05; $\leq$ 7 weeks, ES = 0.29; $p$ = 0.011). The level of certainty of the evidence for the meta-analyzed outcomes ranged from low to moderate.

## Conclusion

Our findings have shown that PT can be effective in enhancing technical skills measures in youth and adult athletes. Sub-group analyses suggest that PT longer (> 7 weeks) lengths appear to be more effective for improving throwing velocity. However, to fully determine the effectiveness of PT in improving sport-specific technical skill outcomes and ultimately enhancing competition performance, further high-quality research covering a wider range of sports is required.

## Introduction

Achievement in sports is generally ascribed to a unique blend of talented and trained physical, technical, tactical, and psychological skills [1]. Evaluating these complex facets may provide valuable information for coaches and trainers regarding the requirements of a particular game or sport and help pinpoint specific aspects that players could improve upon to enhance their performance [2]. Among these factors, a definite correlation exists between technical skills and sports achievement [3]. Sport-specific technical skills refer to actions that involve a specific objective or goal and require the coordination of several motor abilities in a particular context and time frame [4]. Examples of these skills could include passing a soccer ball to a teammate to advance the ball, or throwing a baseball to get an opponent out [2]. The outcome of numerous sports games can be significantly affected by the technical skill level of the participants [3]. For example, in tennis, the serve is regarded as the most crucial technical skill since it can help a player gain an advantage in a point or even secure a direct win [5]. Moreover, it is well known that strength training is commonly associated with performance improvements [6]. Therefore, good training practices to enhance players' technical skills are essential to optimize their performance during matches.

Most coaches and athletes competing at the top levels recognize the value of maintaining and constantly improving technical skill performance [7]. Therefore, an extensive amount of research has been conducted on the development of technical skills employing multiple types of workouts, such as resistance training [8], core training [9], conditioning training [10], and plyometric training (PT) [11]. Notably, PT is a type of strength training that mainly consists of various jumping, hopping, skipping, and throwing exercises [12]. These exercises are inherent in most sports movements, such as high jumping, pitching, or kicking [13, 14]. This method appears to be prevalent [15–18] or even more effective [19, 20] than other training methods (e.g., conventional resistance exercises). The defining feature of PT is the utilization of the stretch-shortening cycle (SSC), which occurs as the muscle rapidly transitions from an eccentric contraction (deceleration phase) to a concentric contraction (acceleration phase) [21]. During SSC tasks, the elastic properties of the muscle and connective tissue are leveraged to store elastic energy during the deceleration phase and release it during the acceleration phase, thus improving the force and power output of the muscle [22, 23]. Plyometrics help build power, a crucial foundation for athletes to improve their sport-specific skills by capitalizing on

the SSC [12]. Moreover, to ensure maximum transfer to sport, the plyometric exercises included in a training program should align with the individual requirements of the athlete and the characteristics of the sport [24]. In other words, the principle of specificity should be followed by selecting plyometric exercises that reflect the type of activity involved in the sport [24]. For instance, some researchers [25, 26] have stated that jumping exercises like vertical jumps were considered irrelevant to improving running performance and therefore did not impact running speed. However, when exercises were designed specifically for running, such as speed bounding, the training program produced a favorable effect on running velocity [27, 28]. Similarly, sprint performance improvements will be optimized by using training regimens that contain greater horizontal acceleration (i.e., jumps with horizontal displacement; sprint-specific plyometric exercises) [24, 29].

Previous reviews and studies have extensively discussed the potential mechanisms (i.e., mechanical and neurophysiological models) implicated in the SSC and its capacity for enhancing human performance [17, 22, 30–33]. The potential benefits and theoretical training improvements associated with plyometric exercises for the upper and lower limbs include, but are not limited to, the following: (a) elevating peak force and acceleration velocity; (b) increasing time for force production; (c) storing energy in the elastic components; (d) increasing muscle activation levels; (e) evoking stretch reflexes [12, 23, 34]. Furthermore, proprioceptors such as the Golgi tendon organ are believed to play a role in a defensive inhibitory reflex response that can be improved through PT. This reflex may elevate performance during activities that involve high-load situations [34, 35]. These adaptations are associated with improvements in several physical qualities, such as strength [36], power [37], agility [38], sprint speed [24], and balance [39]. Indeed, excellent physical ability allows athletes to effortlessly and efficiently execute skills at a high level [40]. For example, to perform gymnastics routines such as vault and tumbling with optimal control and efficiency, gymnasts need to have well-developed explosive power in their upper and lower body muscles [41]. According to Lambrich and Muehlbauer [42], developing optimal agility is crucial for executing tennis-specific footwork and achieving a robust tennis stroke performance. Consequently, in light of the theories mentioned above and the observable improvements in physical characteristics after PT, it seems reasonable to hypothesize that PT could benefit athletes' technical skill performance, regardless of their training backgrounds.

To the best of the authors' knowledge, only one systematic review published in the literature focuses solely on PT's impact on tennis players' technical skill performance [11]. Most reviews focused on the physical fitness aspects of athletes [16, 43, 44]. Additionally, in a previous investigation conducted on soccer players [45], the analysis of technical skill performance as part of physical performance provided a superficial knowledge of the topic and needed to be revised to make significant conclusions. Based on existing reviews, the effects of PT on athletes' technical skill performance still require further clarification. Likewise, a growing number of experimental studies investigated the effects of PT on sports-specific technical skills among athletes. For example, Guadie [46] discovered that PT improves handball team players' technical skill performance in shooting accuracy and speed dribbling. However, this evidence needs to be compiled systematically. Conducting a systematic review with meta-analysis may aid in pinpointing inadequacies and shortcomings in the PT literature and offer practitioners or scholars in related fields valuable insights about prospective future research directions [47]. Therefore, the primary purpose of this systematic review and meta-analysis is to analyze the evidence published about the effects of PT on athletes' technical skill performance and, thus, to expand the current understanding of its effects on athletic populations. For this purpose, this study reviewed experimental trials that compared PT to a comparison group of athletes. All the selected studies met the RCT criteria.

## Methodology

### Protocol and registration

The PRISMA statement [48] was followed in reporting this systematic review and meta-analysis, and the review protocol has been registered on Inplasy.com (INPLASY202320052).

### Eligibility criteria

A PICOS framework [49] was used to rate studies for eligibility. The criteria include the following:

- Population: The participants were healthy female and male athletes of any age and competition level (no restriction).

- Intervention: The minimum PT intervention duration was four weeks. Plyometric exercise programs can focus on either the upper or lower limbs or a combination. These exercises may include medicine ball exercises, plyometric push-ups, bilateral or unilateral bounds, hops, jumps, and skips. These exercises typically involve a pre-stretch or countermovement that stresses the SSC. PT combined with other types of strength training (e.g., weight training) was excluded to avoid confounding effects [37]; however, trials that included combined training were included if it was specified that the control group received the same training as the experimental group, except for the PT component.

- Control: The control group was in a regular sports program without plyometric exercises.

- Outcomes: The study's results need to include the effect of at least one plyometric exercise on the technical skill performance of participants. A technical skill outcome was defined as actions that involve a specific task or goal and require the coordination of several motor abilities in a particular context and time frame [4]. To be eligible for inclusion, studies needed to report the treatment effect or pre- and post-test values for technical skill outcomes specific to the sport. Studies that solely assessed time trial performance outcomes (e.g., running) rather than sport-specific technical skills (e.g., stride frequency) were not considered.

- Study design: This review considered randomized controlled trials.

    Furthermore, studies that did not meet the inclusion criteria were rejected from this review.

### Search strategy and selection process

On the 3rd of May 2023, the following four electronic databases were searched to obtain articles pertinent to the topic: Web of Science Core Collection, SPORTDiscus, PubMed, and SCOPUS. Previous reviews [50, 51] were used to help define our search strategy; keywords and Boolean operators were considered separately and in aggregation while searching the four databases (S1 Table). This study employed the following terms and operators: ("plyometric training" OR "plyometric exercise*" OR "stretch-shortening cycle" OR "jump training") AND ("athletic performance" OR "technical skill*" OR "skill*" OR "technique" OR "performance") AND (athlete* OR player*). Moreover, to find studies that could potentially be incorporated into this systematic review, we scrutinized pertinent review articles that were published prior to May 3, 2023 [11, 45, 47, 52, 53]. Furthermore, relevant supplementary material was searched for through manual searches on Google Scholar, including article citations and free-text searches. In addition, we screened the reference lists of all the identified articles to discover any publications that were not detected by the initial computerized search. Finally, experienced librarians backed up the data-gathering process and ensured that the process was performed correctly.

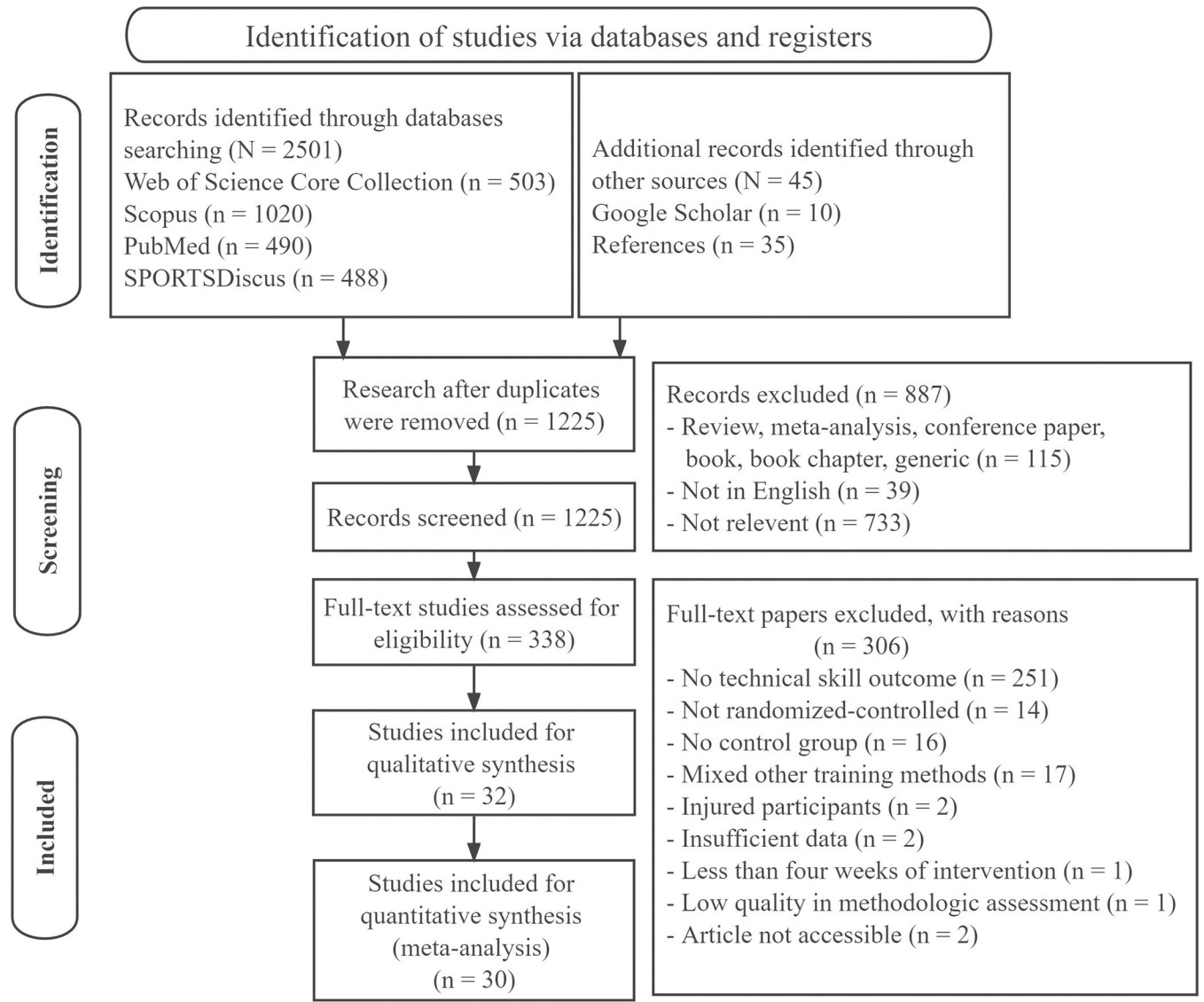

**Fig 1. PRISMA flow diagram.**

The study selection included four significant processes (Fig 1). Initially, duplicate articles were eliminated, and the title and abstract were determined in the second stage. Studies were excluded if they dealt with different subjects or were written in a language other than English. The review only included articles written in English because articles written in other languages can be challenging to translate. Moreover, almost all of the literature (99.6%) on plyometric jump training is published in English [54]. Conference abstracts, books, book sections, pilot studies, or papers not published in peer-review journals were excluded. The full-text screening process included a review of predetermined eligibility criteria. Furthermore, studies were excluded if they inadequately reported PT, if testing protocols for technical skill performance measurements were not conducted under supervision, if the entire text was unavailable from the database or the authors. Two independent reviewers (ND and DH) completed this process. Any dispute was further explored. If necessary, a third reviewer (KGS) helped until an agreement was obtained.

## Data extraction

Two reviewers (ND and DH) obtained information about each study by utilizing a Microsoft Excel spreadsheet (Microsoft Corporation, Redmond, WA, United States), and a third reviewer (KGS) verified its accuracy. The records taken into account were: a) name of the first author, year, and country of publication; b) subject characteristics: age, sex, sample size, competition level, and sports experience; and c) characteristics of the PT intervention, which include training length, frequency, time, type of exercise, PT replaced (if applicable) a component of the regular practice; d) assessment of technical skill performance: several measures of technical skills were selected after discussions among the co-authors, including, but not limited to: kicking (e.g., soccer), dribbling (e.g., handball), passing (e.g., basketball), serving (e.g., tennis); e) mean and standard deviation of the results for PT and control groups.

## Study quality assessment

Two authors (ND, KGS) independently utilized the PEDro scale, which is a valid [55] and reliable [56] instrument for grading the methodological quality of investigations. The results were cross-checked by the third author (AB) and all three reviewers achieved agreement. On a scale of 1 to 10, $\leq$ 3 points indicated poor quality, 4–5 points indicated moderate quality, and 6–10 points indicated high quality. The PEDro scale comprises 11 elements used to evaluate methodological quality. Each fulfilled item provides one point to the total PEDro score (0–10 points). Regarding external validity, criterion 1 was excluded from the research quality evaluation. To decrease the possibility of a high risk of bias in the analysis, it was decided after the fact to exclude studies that were rated with a score of 3 or lower. A discussion with a third reviewer (BA) resolved disagreements between the two reviewers.

## Certainty of evidence

The certainty of the evidence was evaluated by two authors (ND and DH) using the Grading of Recommendations, Assessment, Development and Evaluation (GRADE) methodology, which categorized it as very low, low, moderate, or high [57–59]. The evidence was initially rated as high for each outcome, but was later downgraded after evaluating the following criteria: (a) risk of bias in studies: if the median PEDro scores were moderate (less than 6), the judgments were lowered by one level; (b) indirectness: low risk of indirectness was ascribed by default owing to the specificity of populations, interventions, comparators, and outcomes being ensured by the inclusion/exclusion criteria; (c) risk of publication bias: downgraded by one level if there was suspected publication bias; (d) inconsistency: judgments were downgraded by one level when the impact of statistical heterogeneity ($I^2$) was high ($> 75\%$); (e) imprecision: one level of downgrading occurred whenever $< 800$ participants were available for a comparison [60] and/or if the effects' direction was unclear. When both were observed, certainty was downgraded by two levels. If the number of comparison trials was insufficient to conduct a meta-analysis, the evidence was automatically considered of very low certainty [47]. Therefore, for the outcomes not included in the meta-analyses, the certainty of evidence should be regarded as very low.

## Statistical analysis

Although a meta-analytical comparison can be made with just two studies [61], the field of PT often has small sample sizes [47]. Therefore, we only conducted meta-analyses when three or more studies reported data on the technical skill outcomes mentioned earlier [11]. To calculate effect sizes (ES) (Hedges' g), means and standard deviations of a measure of pre-post-intervention performance were utilized, and the data were standardized using the post-intervention data of a

relevant performance measure. If data values were not accessible, such as when they were omitted or displayed in graphical form, the corresponding author of the study was contacted to obtain the necessary information. In cases where data were presented graphically without numerical data, validated (r = 0.99, p < 0.001) [62] software (WebPlot- Digitizer, version 4.5; https://apps.automeris.io/wpd/) was employed to extract the numerical data from the figures.

The meta-analyses employed the inverse variance random-effects model, which assigns weights to trials proportional to their individual standard errors [63], and facilitates analysis while accounting for heterogeneity across studies [64]. The ESs were presented with 95% confidence intervals (95% CIs), and standardized mean differences were used to interpret the calculated ESs, with conventions as follows: <0.2, trivial; 0.2–0.6, small; >0.6–1.2, moderate; >1.2–2.0, large; >2.0–4.0, very large; >4.0, extremely large [65]. In studies with multiple intervention groups, the control group was proportionally divided to enable comparison across all participants [66]. The $I^2$ statistics were used to assess the impact of study heterogeneity, with values of <25%, 25–75%, and >75% indicating low, moderate, and high levels of heterogeneity, respectively [67]. The extended Egger's test was used to investigate the risk of publication bias [68], and a sensitivity analysis was performed in the case of a significant Egger's test. All analyses were performed using the Comprehensive Meta-Analysis program (version 3; Biostat, Englewood, NJ, USA), with a statistical significance threshold of p < 0.05.

### Additional analysis

Subgroup analyses were conducted to examine the potential impact of moderator factors. Relevant sources of heterogeneity that could affect the training effects were pre-selected based on the authors' discussion and study characteristics: program length (weeks), training frequency (sessions per week), the total number of training sessions, and weekly session time. Participants were divided using a median split [30, 69] for training length (i.e., ≤ 7 vs. > 7 weeks), the total number of PT sessions (i.e., ≤ 14 vs. > 14 sessions), and the time of PT session itself (main part) (i.e., ≤ 30 vs. > 30 minutes). As the majority of studies utilized a training frequency of 2 or 3 sessions per week (see Table 3), we grouped the training frequency as either 2 or 3 sessions per week to facilitate comparison. For each moderator factor, the median was computed if there were at least three studies providing data. Additionally, we examined the gender (male vs. female) and age (< 18 years of age vs. ≥ 18 years of age) of the athletes as potential moderator factors.

## Results

### Study selection

Fig 1 illustrates that a total of 2501 papers were initially found through searches on databases. An additional 45 studies were identified through Google Scholar and reference lists. After manually removing duplicates, 1225 records remained. The titles and abstracts of the 1225 records were checked. After screening the titles and abstracts of articles, 338 papers were identified as potentially eligible for full-text analysis. However, after conducting a full-text examination, 306 publications were excluded. Ultimately, 32 papers met the inclusion criteria and 30 of those were eligible for meta-analysis.

### Study quality assessment

The PEDro checklist was used to assess each paper's quality, and the results showed that seven studies received a score of 4 or 5, indicating a moderate quality. Meanwhile, 25 studies scored 6 to 9 points, considered highly methodological. However, a study [70] with a quality score of ≤ 3 points was noted and thus excluded from this review (Table 1).

**Table 1. Physiotherapy Evidence Database (PEDro) scale ratings.**

| Study name | N° 1 | N° 2 | N° 3 | N° 4 | N° 5 | N° 6 | N° 7 | N° 8 | N° 9 | N° 10 | N° 11 | Total* | Study quality |
|---|---|---|---|---|---|---|---|---|---|---|---|---|---|
| Behringer et al. [19] | 1 | 1 | 1 | 1 | 0 | 0 | 0 | 1 | 1 | 1 | 1 | 7 | High |
| Guadie et al. [46] | 0 | 1 | 0 | 1 | 0 | 0 | 0 | 0 | 0 | 1 | 1 | 4 | Moderate |
| Lee et al. [70] | 0 | 1 | 0 | 0 | 0 | 0 | 0 | 0 | 0 | 1 | 1 | 3 | Low |
| Saunders et al. [71] | 1 | 1 | 0 | 1 | 0 | 0 | 0 | 1 | 1 | 1 | 1 | 6 | High |
| Carter et al. [72] | 0 | 1 | 0 | 1 | 0 | 0 | 0 | 1 | 1 | 1 | 1 | 6 | High |
| Bishop et al. [73] | 1 | 1 | 0 | 1 | 0 | 0 | 0 | 1 | 1 | 1 | 1 | 6 | High |
| Campo et al. [74] | 0 | 1 | 0 | 1 | 0 | 0 | 0 | 1 | 0 | 1 | 1 | 5 | Moderate |
| Sedano et al. [75] | 1 | 1 | 1 | 1 | 0 | 0 | 0 | 1 | 1 | 1 | 1 | 7 | High |
| Escamilla et al. [76] | 1 | 1 | 0 | 1 | 0 | 0 | 0 | 1 | 1 | 0 | 1 | 5 | Moderate |
| Sharma and Multani [77] | 0 | 1 | 0 | 1 | 0 | 0 | 0 | 0 | 0 | 1 | 1 | 4 | Moderate |
| Michailidis et al. [78] | 0 | 1 | 0 | 1 | 0 | 0 | 0 | 1 | 1 | 1 | 1 | 6 | High |
| Ramírez-Campillo et al. [79] | 1 | 1 | 1 | 1 | 0 | 0 | 0 | 1 | 1 | 1 | 1 | 7 | High |
| Ramírez-Campillo et al. [80] | 1 | 1 | 0 | 1 | 0 | 0 | 0 | 1 | 1 | 1 | 1 | 6 | High |
| Chelly et al. [81] | 1 | 1 | 0 | 1 | 0 | 0 | 0 | 1 | 1 | 1 | 1 | 6 | High |
| De Villarreal et al. [82] | 1 | 1 | 0 | 1 | 0 | 0 | 0 | 1 | 1 | 1 | 1 | 6 | High |
| Ramı́rez-Campillo et al. [83] | 1 | 1 | 0 | 1 | 0 | 0 | 0 | 1 | 1 | 1 | 1 | 6 | High |
| Ramı́rez-Campillo et al. [84] | 1 | 1 | 0 | 1 | 0 | 0 | 0 | 1 | 1 | 1 | 1 | 6 | High |
| Ramı́rez-Campillo et al. [85] | 1 | 1 | 1 | 1 | 0 | 0 | 1 | 1 | 1 | 1 | 1 | 8 | High |
| Ramos-Veliz et al. [86] | 1 | 1 | 0 | 1 | 0 | 0 | 0 | 1 | 1 | 1 | 1 | 6 | High |
| De Villarreal et al. [87] | 1 | 1 | 0 | 1 | 0 | 0 | 0 | 1 | 1 | 1 | 1 | 6 | High |
| Hall et al. [88] | 1 | 1 | 0 | 1 | 0 | 0 | 0 | 1 | 1 | 1 | 1 | 6 | High |
| Giovanelli et al. [89] | 1 | 1 | 1 | 1 | 0 | 0 | 0 | 1 | 1 | 1 | 1 | 7 | High |
| Ache-Dias et al. [90] | 1 | 1 | 0 | 1 | 0 | 0 | 0 | 1 | 1 | 1 | 1 | 6 | High |
| Ramirez-Campillo et al. [91] | 1 | 1 | 1 | 1 | 0 | 0 | 1 | 1 | 1 | 1 | 1 | 8 | High |
| Ramirez-Campillo et al. [92] | 1 | 1 | 1 | 1 | 0 | 0 | 0 | 1 | 1 | 1 | 1 | 7 | High |
| Gómez-Molina et al. [93] | 1 | 1 | 0 | 1 | 0 | 0 | 0 | 1 | 1 | 1 | 1 | 6 | High |
| Ramírez-Campillo et al. [94] | 1 | 1 | 0 | 1 | 0 | 1 | 0 | 1 | 1 | 1 | 1 | 7 | High |
| Ramirez-Campillo et al. [95] | 1 | 1 | 1 | 1 | 0 | 1 | 1 | 1 | 1 | 1 | 1 | 9 | High |
| Vera-Assaoka et al. [96] | 1 | 1 | 0 | 1 | 0 | 0 | 1 | 1 | 1 | 1 | 1 | 7 | High |
| Aloui et al. [97] | 0 | 1 | 0 | 1 | 0 | 0 | 0 | 1 | 0 | 1 | 1 | 5 | Moderate |
| Alp and Ozdinc [98] | 0 | 1 | 0 | 1 | 0 | 0 | 0 | 0 | 0 | 1 | 1 | 4 | Moderate |
| Çalışkan and Arikan [99] | 0 | 1 | 0 | 1 | 0 | 0 | 0 | 0 | 0 | 1 | 1 | 4 | Moderate |
| Wee et al. [100] | 0 | 1 | 0 | 1 | 0 | 0 | 0 | 1 | 1 | 1 | 1 | 6 | High |

Note: A detailed explanation for each PEDro scale item can be accessed at https://www.pedro.org.au/english/downloads/pedro-scale.

*From a possible maximal punctuation of 10.

## Certainty of evidence

Table 2 presents the GRADE analyses, which indicate that the certainty of the evidence was low to moderate for all outcomes and group comparisons. In addition, for comparisons that were not analyzed using meta-analysis, the evidence was suggested to be of very low certainty.

## Study characteristics

Table 3 displays the participant and intervention characteristics of the RCTs included in this review. All research that satisfied the criteria for inclusion was released in English from 2006 to 2023. Out of the 32 articles analyzed, 21 were two-armed trials [46, 71–75, 77–79, 81, 82, 86,

**Table 2. Certainty of evidence for meta-analyzed outcomes.**

| Outcomes | Certainty assessment | | | | | No of participants and studies | Certainty of evidence (GRADE) |
|---|---|---|---|---|---|---|---|
| | Risk of bias | Inconsistency | Indirectness | Imprecision | Risk of publication bias | | |
| Technical skill performance assessed with: kicking velocity follow-up: range 6 to 12 weeks | Not Serious | Not serious | Not serious | Serious [c] | Serious [e] | 267 (8 RCTs) | ⊕⊕◯◯LOW |
| Technical skill performance assessed with: throwing velocity follow-up: range 6 to 16 weeks | Serious [a] | Not serious | Not serious | Serious [c] | Not serious | 270 (6 RCTs) | ⊕⊕◯◯LOW |
| Technical skill assessed with: kicking distance follow-up: range 7 to 12 weeks | Not Serious | Not serious | Not serious | Serious [c] | Not serious | 363 (6 RCTs) | ⊕⊕⊕◯ MODERATE |
| Technical skill performance assessed with: speed dribbling follow-up: range 4 to 12 weeks | Serious [a] | Not Serious | Not serious | Serious [c] | Not serious | 87 (3 RCTs) | ⊕⊕◯◯LOW |
| Technical skill performance assessed with: stride rate follow-up: range 4 to 12 weeks | Not serious | Not Serious | Not serious | Serious [c] | Not serious | 91 (4 RCTs) | ⊕⊕⊕◯ MODERATE |

GRADE, Grading of Recommendations Assessment, Development and Evaluation

a Downgraded by one level due to average PEDro score being moderate ($< 6$)

b Downgraded by one level due to high impact of statistical heterogeneity ($> 75\%$)

c Downgraded by one level, as $\geq 800$ participants were available for a comparison or there was an unclear direction of the effects

e Downgraded by one level (Egger's test $< 0.05$); GRADE Working Group grades of evidence High certainty: we are very confident that the true effect lies close to that of the estimate of the effect. Moderate certainty: we are moderately confident in the effect estimate: the true effect is likely to be close to the estimate of the effect, but there is a possibility that it is substantially different. Low certainty: our confidence in the effect estimate is limited: the true effect may be substantially different from the estimate of the effect. Very low certainty: we have very little confidence in the effect estimate: the true effect is likely to be substantially different from the estimate of effect

88–90, 93, 94, 97–100], four were three-armed [19, 87, 91, 95], and the remaining four had four arms [76, 80, 83, 84]. Thirteen RCTs were conducted in America (ten in Chile, one in Brazil, and two in the United States) [72, 76, 79, 80, 83–85, 90–92, 94–96], eleven in Europe (three in Spain, one in Ethiopia, one in Germany, one in Greece, one in France, one in Italy, one in Turkey, and two in the United Kingdom) [19, 46, 73, 78, 86–89, 93, 98, 99]. Two RCTs were conducted in Asia (one in Malaysia; one in India) [77, 100], three took place in Africa (two in Tunisia, one in Ethiopia) [46, 97, 81], and one in Oceania (Australia) [71].

Out of the 32 documents, sixteen focused on soccer players [74, 75, 78–80, 83–85, 91, 92, 94–96, 99–100]; four articles on handball players [46, 81, 97, 98]; three articles on runners [71, 89, 90]; three articles on water polo players [82, 86, 87]; two studies on baseball players [72, 76]; and the remaining four articles were on swimmers [73], basketball players [77], gymnasts [88], tennis players [19]. There were 1078 participants in total, with 644 men (59.7%) and 84 women (7.8%), while 350 (32.5%) did not report gender or mixed gender. The mean age of the participants ranged from 10 to 40 years old. In terms of athletes' expertise level, eight studies employed regional or local sport club athletes; ten studies were categorized as national players. Three studies recruited university or college players, two studies selected high school or youth soccer school athletes, and two studies recruited amateur or recreational athletes. However, seven studies did not report participant levels. In addition, 12 studies reported between two months and eight years of specific-sport experience, whereas ten studies did not report the duration of participants' sport-specific experience.

Different PT modes were employed in the experimental groups, most of which were lower limb plyometric exercises (n = 21). Four studies used upper limb plyometric training, and

**Table 3. Characteristics of participants and PT interventions in the included studies.**

| Study | Country | Intervention | N | Sex | Age | Level/ Experience | Sport | Comparison | Replace | Outcome (s) |
|---|---|---|---|---|---|---|---|---|---|---|
| Behringer et al. [19] | Germany | Freq: 2 times/ week Time: 45 min Length: 8 weeks | 36 | M | 15.03 ± 1.64 yrs | Local clubs averaged 6.15 yrs | Tennis | EG1:Resistance-training EG2:ULLPT CG: Regular tennis training | No | Serve velocity↑ EG2 > EG1 Serve accuracy→ EG1 = EG2 |
| Guadie. [46] | Ethiopia | Freq: 3 times/ week Time: 60 min Length: 12 weeks | 22 | M | NR | Regional NR | Handball | EG:ULLPT CG:Regular handball practice | No | Shooting accuracy ↑ Speed dribbling ↑ Passing accuracy→ |
| Saunders et al. [71] | Australia | Freq: 2–3 times/ week Time: 30 min Length: 9 weeks | 15 | M | EG: 23.4 ± 3.2 yrs CG: 24.9 ± 3.2 yrs | National level NR | Running | EG: LLPT CG: Usual running training | No | Stride rate ↑ |
| Carter et al. [72] | United States | Freq: 2 times/ week Time: NR Length: 8 weeks | 24 | M | 19.7± 1.3 yrs | Collegiate NR | Baseball | EG: ULPT CG: Routine training | No | Throwing velocity ↑ |
| Bishop et al. [73] | United Kingdom | Freq: 2 times/ week Time: NR Length: 8 weeks | 22 | NR | 10–16 yrs | Local clubs NR | Swimming | EG:LLPT CG: Habitual training | No | Swimming block start ($T_{head\ contact}$ ↑, $A_{blocks} \rightarrow A_{water}$ →, $V_{take\ off}$ ↑) |
| Campo et al. [74] | Spain | Freq: 3 times/ week Time: NR Length: 12 weeks | 20 | F | CG: 23.0 ± 3.2 yrs; EG: 22.8 ± 2.1 yrs | National 5.2 6 ± 3.2 yrs | Soccer | EG: LLPT CG: Regular soccer training | Yes | Kicking velocity ↑ |
| Sedano et al. [75] | Spain | Freq: 3 times/ week Time: NR Length: 10 weeks | 22 | NR | CG18.2±0.9 yrs EG18.4±1.1 yrs | National NR | Soccer | EG: LLPT CG: Normal soccer training | Yes | Kicking velocity↑ |
| Escamilla et al. [76] | United States | Freq: 3 times/ week Time: 45 min Length: 6 weeks | 68 | NR | 14–17 yrs | High school NR | Baseball | EG1:Throwers Ten EG2:Keiser Pneumatic EG3: ULPT CG: Summer baseball activity | No | Throwing velocity ↑ EG1 = EG2 = EG3 |
| Sharma and Multani [77] | India | Freq: 2 times/ week Time: NR Length: 4 weeks | 40 | M | 12–20 | NR NR | Basketball | EG:ULLPT CG: Regular basketball practice | No | Passing accuracy↑ Speed dribbling ↑ |
| Michailidis et al. [78] | Greece | Freq: 2 times/ week Time: 20–25 min Length: 12 weeks | 45 | M | 10.6 ± 0.6 yrs 10.6 ± 0.5 yrs | NR 3.4 ± 0.4 yrs 3.6 ± 0.6 yrs | Soccer | EG2: LLPT CG: Regular soccer training | No | Kicking distance↑ |
| Ramírez-Campillo et al. [79] | Chile | Freq: 2 times/ week Time: 21 min Length: 7 weeks | 76 | M | 13.2 ± 1.8 | NR > 2 yrs | Soccer | EG: LLPT CG: Normal soccer training | Yes | Kicking distance ↑ |
| Ramírez-Campillo et al. [80] | Chile | Freq: 2 times/ week Time: 15–25 min Length: 7 weeks | 54 | M | 10.4 ± 2.3 yrs | Amateur soccer team 3.3 ± 1.5 yrs | Soccer | EG1: LLPT (30s rest interval) EG2: LLPT (60s rest interval) EG3: LLPT (120s rest interval) CG: Regular soccer training | Yes | Kicking distance ↑ |
| Chelly et al. [81] | Tunisia | Freq: 2 times/ week Time: 30 min Length: 8 weeks | 23 | M | 17.2 ± 0.4 yrs | National 7.2 6 ±1.1yrs | Handball | EG:ULLPT CG: Standard regimen | Yes | Throwing velocity ↑ |

(*Continued*)

**Table 3.** (Continued)

| Study | Country | Intervention | N | Sex | Age | Level/Experience | Sport | Comparison | Replace | Outcome (s) |
|---|---|---|---|---|---|---|---|---|---|---|
| De Villarreal et al. [82] | Spain | Freq: 3 times/week Time: 45 min Length: 6 weeks | 19 | M | EG1 18.5 ± 2.3 yrs EG2 19.7 ± 5.4 yrs | National EG1 10.5 ± 2.1 yrs EG2 11.5 ± 4.1 yrs | Water polo | EG:ULLPT CG:in-water strength training | No | Throwing velocity → |
| Ramı´rez-Campillo et al. [83] | Chile | Freq: 2 times/week Time: 40 min Length: 6 weeks | 54 | M | 11.4 ± 2.2 yrs | Sub-elite level 3–4 yrs | Soccer | EG1: LLPT (bilateral jumps) EG2: LLPT (unilateral jumps) EG3: LLPT (bilateral + unilateral jumps) CG: Regular soccer training | Yes | Kicking velocity ↑ |
| Ramı´rez-Campillo et al. [84] | Chile | Freq: 2 times/week Time: 40 min Length: 6 weeks | 40 | M | 10–14yrs | NR EG1:3.6 ± 2.3yrs EG2: 4.1 ± 2.6yrs EG3: 3.5 ± 2.3yrs CG: 3.9 ± 2.3yrs | Soccer | EG1: LLPT (vertical jumps) EG2: LLPT (horizontal jumps) EG3: LLPT (vertical + horizontal jumps) CG: Regular soccer training | Yes | Kicking velocity ↑ EG3> EG1,EG2 |
| Ramı´rez-Campillo et al. [85] | Chile | Freq: 2 times/week Time: 40 min Length: 6 weeks | 24 | M | 13.0 ± 2.3 yrs | Local soccer club EG1: 4.0 ± 1.4 yrs EG2: 4.1 ± 1.5yrs CG: 4.1 ± 1.5yrs | Soccer | EG1: LLPT with a progressive increase in volume EG2: LLPT without a progressive increase in volume CG: Regular soccer training | Yes | Kicking velocity EG1> EG2 |
| Ramos-Veliz et al. [86] | Spain | Freq: 2 times/week Time: 45 min Length: 16 weeks | 21 | F | 26.4 ± 4.3 yrs | National 10.6 ± 4.1 yrs | Water polo | EG:LLPT CG:In-water strength training | No | Throwing velocity ↑ |
| De Villarreal et al. [87] | Spain | Freq: 3 times/week Time: 45 min Length: 6 weeks | 30 | M | 23.4 ± 4.1 yrs | National 7.8 ± 3.1 yrs | Water polo | EG1: In-water strength training EG2: ULLPT CG: dry-land + in-water strength training | No | Throwing velocity ↑ |
| Hall et al. [88] | United Kingdom | Freq: 2 times/week Time: 45 min Length: 6 weeks | 20 | F | 12.5 ± 1.67yrs | Local club ≥ 3 yrs | Gymnastic | EG:ULLPT CG: Habitual gymnastic training | No | Handspring vault ($V_{run\text{-}up}$ ↑*, $V_{take\,off}$ ↑ HBD ↑, BCT ↑, TCT →, Pre-FT →, Post-FT→) |
| Giovanelli et al. [89] | Italy | Freq: 3 times/week Time: 25–30 min Length:12 weeks | 25 | M | 38.2 ± 7.1 yrs | Well-trained & national level 11.7 ± 8.6 yrs & 4.7 ± 3.4 yrs | Running | EG: LLPT CG: Usual running training | No | Stride rate → Stride length → |
| Ache-Dias et al. [90] | Brazil | Freq: 2 times/week Time: NR Length:4 weeks | 26 | Mixed | 18–40 yrs | Recreational ≥ 2 months | Running | EG: LLPT CG: Usual running training | No | Stride rate ↑ Stride length ↑ |
| Ramirez-Campillo et al. [91] | Chile | Freq:1/2times/week Time: 6–20 min Length: 8 weeks | 23 | F | 21.4 ± 3.2 yrs | Amateur & Regional EG1:5.4 ±1.4 yrs EG2:5.6 ±1.8 yrs CG:6.0 ±1.6 yrs | Soccer | EG1: One session LLPT EG2: Two sessions LLPT CG: Regular soccer training | Yes | Kicking velocity ↑ EG1> EG2 |

(*Continued*)

**Table 3.** (Continued)

| Study | Country | Intervention | N | Sex | Age | Level/ Experience | Sport | Comparison | Replace | Outcome (s) |
|---|---|---|---|---|---|---|---|---|---|---|
| Ramirez-Campillo et al. [92] | Chile | Freq: 2 times/ week Time: 10–17 min Length: 7 weeks | 73 | M | 10.9–15.9 yrs | National NR | Soccer | EG1:LLPT using a fixed drop-box height EG2: LLPT using a optimal drop-box height CG: Regular soccer training | Yes | Kicking distance → |
| Gómez-Molina et al. [93] | Spain | Freq: 2 times/ week Time: 25–45 min Length: 8 weeks | 25 | M | Novice EG: 20.4 ± 2.4 yrs CG: 20.7 ± 1.8 yrs | NR NR | Running | EG: LLPT CG: Usual running training | No | Stride rate ↑ |
| Ramírez-Campillo et al. [94] | Chile | Freq: 2 times/ week Time: 20 min Length: 7 weeks | 39 | M | EG:13.2 ± 1.8 yrs CG:13.5 ± 1.9 yrs | NR > 2-yrs | Soccer | EG: LLPT CG: Regular soccer training | Yes | Kicking distance↑ |
| Ramirez-Campillo et al. [95] | Chile | Freq: 2 times/ week Time: 10–15 min Length: 8 weeks | 23 | M | 11–14 yrs | NR EG1:3.8 ± 1.4 yrs EG2: 3.8 ± 1.4 yrs CG: 4.0 ± 1.6 yrs | Soccer | EG1: Combined surfaces LLPT EG2: Single-surface LLPT CG: Regular soccer training | Yes | Kicking velocity ↑ EG1> EG2 |
| Vera-Assaoka et al. [96] | Chile | Freq: 2 times/ week Time: 21 min Length: 7 weeks | 76 | M | EG1: 11.2 ± 0.8 yrs EG2: 14.4 ± 1.0 yrs CG1: 11.5 ± 0.9 yrs CG2: 14.5 ± 1.1 yrs | Local soccer team EG1: 3.3 ± 0.9 yrs EG2: 5.4 ± 1.9 yrs CG1: 3.7 ± 1.0 yrs CG2: 5.1 ± 2.0 yrs | Soccer | EG1:LLPT EG2: LLPT CG1:Regular handball practice CG2: Regular handball practice | No | Kicking distance → |
| Aloui et al. [97] | Tunisia | Freq: 2 times/ week Time: 20 min Length: 8 weeks | 29 | NR | 17.7 ± 0.4 yrs | National 6.3 ± 0.8 yrs | Handball | EG: Elastic band ULPT CG: Technical-tactical training | Yes | Throwing Velocity ↑ |
| Alp and Ozdinc [98] | Turkey | Freq: 3 times/ week Time: 30 min Length: 8 weeks | 20 | M | CG 20.60 ± 1.35 yrs EG 22.10 ± 2.13 yrs | University NR | Handball | EG: ULPT CG: Routine training | No | Throwing velocity ↑ |
| Çalışkan and Arikan [99] | Turkey | Freq: 2 times/ week Time: 30 min Length: 8 weeks | 25 | M | 13–15 yrs | Youth Soccer School 1 yr | Soccer | EG: LLPT CG: Regular soccer training | No | Speed dribbling ↑ |
| Wee et al. [100] | Malaysia | Freq: 2 times/ week Time: NR Length: 6 weeks | 19 | M | EG: 18.6±0.7 yrs CG: 20.0±1.4 yrs | Collegiate EG: 4.7±2.9 yrs CG: 8.1±2.4 yrs | Soccer | EG: LLPT CG: Regular soccer training | No | Kicking velocity ↑ |

F, female; M, male; yrs, years; Freq, frequency; NR, not reported; EG, experimental group; CG, control group; Replace, replacement of a portion of the habitual training drills with plyometric training drills; ULPT, upper limb plyometric training; LLPT, lower limb plyometric training; ULLPT, combined upper and lower limb plyometric training; $T_{head\ contact}$, time to head contact, $A_{blocks}$, sangle out of blocks; $A_{water}$, angle of entry into water; $V_{run-up}$, run-up velocity; $V_{take\ off}$, take-off velocity; HBD, hurdle to board distance; BCT, board contact time; TCT, table contact time; Pre-FT, pre-flight time; Post-FT, post-flight time; ↑, significant within-group improvement; →, no significant within-group improvement.

seven studies employed combined upper and lower limb plyometric training exercises. The average length of PT was 7.9 weeks, with a range of 4 to 16 weeks, and the average frequency per week was 2.4 sessions, ranging from 1 to 3 sessions. Out of the included studies, 14 reported session durations between 10 and 30 minutes, and 11 reported between 30 and 60 minutes. Seven papers did not provide information on the duration of the sessions.

## Meta-analysis results

The meta-analysis focused on 30 studies that evaluated athletes' technical skills, specifically measuring kicking velocity and distance (in soccer), throwing velocity (in handball, baseball, and tennis), dribbling speed (in soccer and handball), and stride rate (in running). The data used for the meta-analyses can be found in S2 Table.

Six studies provided data for throwing velocity, involving nine experimental and six control groups (pooled n = 270). Results showed a moderate effect of PT on kicking velocity (ES = 0.78; 95% CI = 0.49–1.07; $p < 0.001$; $I^2 = 8.7\%$; Egger's test $p = 0.15$; Fig 2). The weight value of every study ranged from 8.94 to 14.88% in the analysis.

Eight studies provided data for kicking velocity, involving 15 experimental and eight control groups (pooled n = 267). The Egger's test revealed a $p = 0.008$. After sensitivity analysis, the removal of one study [74] allowed an Egger's test $p > 0.05$. As such, 14 studies with seven experimental and three control groups were finally considered. There was a small effect of PT on kicking velocity performance (ES = 0.37; 95% CI = 0.08–0.65; $p = 0.011$; $I^2 = 0.0\%$; Egger's test $p = 0.072$; Fig 3). The weight value of every study ranged from 5.21 to 11.55% in the analysis.

Six studies provided data for kicking distance, involving ten experimental and six control groups (pooled n = 363). Results showed a small effect of PT on kicking distance (ES = 0.44; 95% CI = 0.23–0.65; $p < 0.001$; $I^2 = 0.0\%$; Egger's test $p = 0.970$; Fig 4). The weight value of every study ranged from 3.81 to 22.18% in the analysis.

Three studies provided data for speed dribbling, involving three experimental and three control groups (pooled n = 87). Results showed a moderate effect of PT on speed dribbling

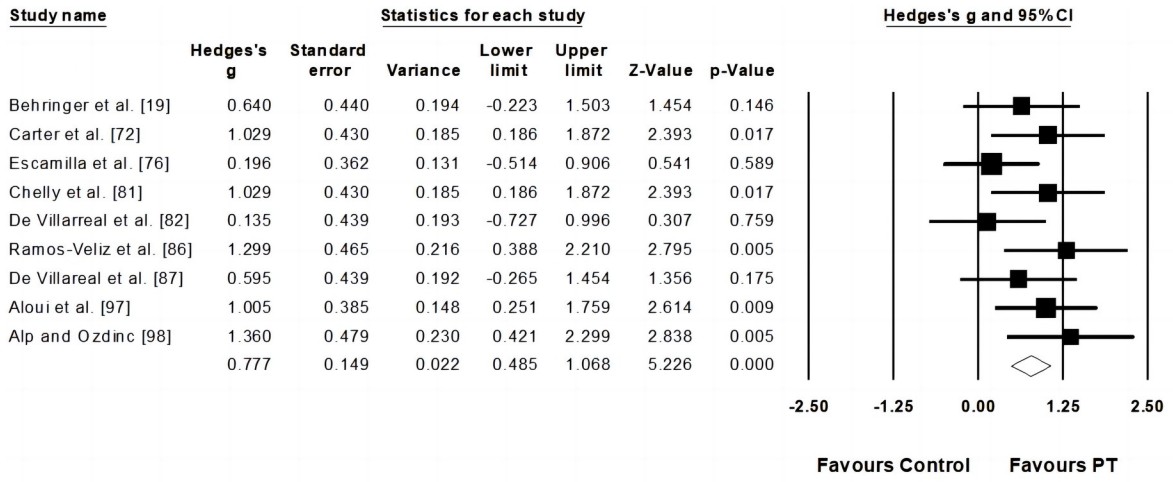

**Fig 2. Forest plot of changes in throwing velocity performance in athletes participating in plyometric training compared to controls.** Values shown are effect sizes (Hedges's g) with 95% confidence intervals (CI). The size of the plotted squares reflects the statistical weight of the study.

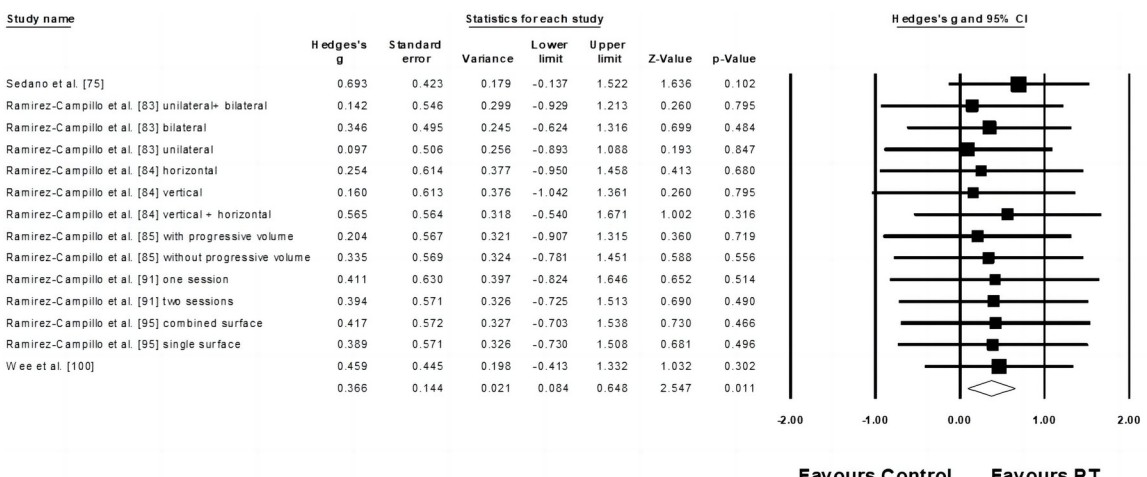

**Fig 3. Forest plot of changes in kicking velocity performance in athletes participating in plyometric training compared to controls.**
Values shown are effect sizes (Hedges's g) with 95% confidence intervals (CI). The size of the plotted squares reflects the statistical weight of the study.

(ES = 0.85; 95% CI = -0.17–1.52; p = 0.014; $I^2$ = 57.6%; Egger's test $p$ = 0.799; Fig 5). The weight value of every study ranged from 25.40 to 42.03% in the analysis.

Four studies provided data for stride rate performance, involving four experimental and four control groups (pooled n = 91). Results showed a small effect of PT on kicking velocity (ES = 0.32; 95% CI = -0.10–0.73; p = 0.137; $I^2$ = 0.0%; Egger's test $p$ = 0.393; Fig 6). The weight value of every study ranged from 19.08 to 29.82% in the analysis.

## Additional analysis

Due to a restricted number of trials (three per moderator), only 16 analyses of moderators were performed (as shown below).

Regarding subject-related moderator variables, when compared to younger athletes, no significant increases were observed following PT in their older counterparts, for kicking velocity

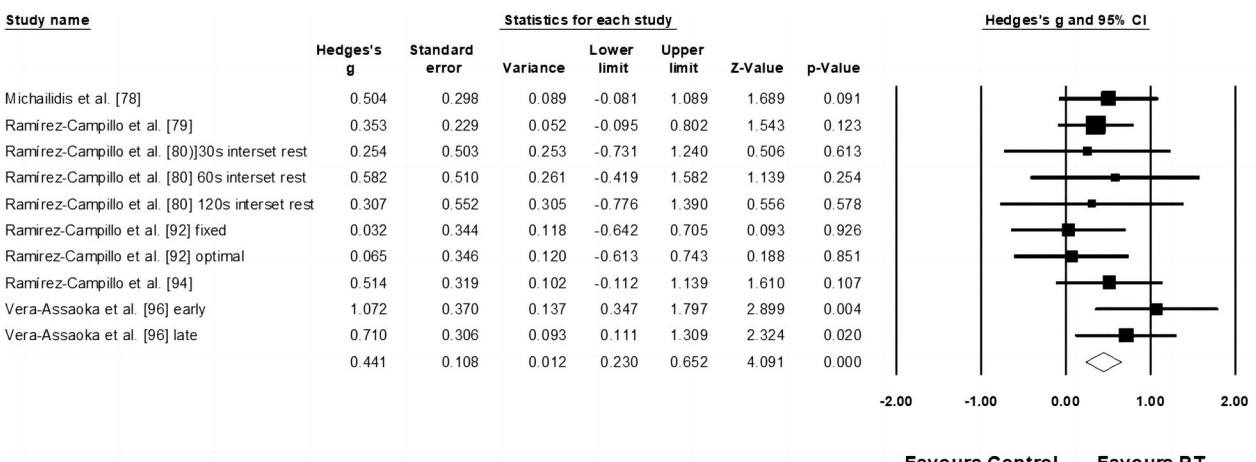

**Fig 4. Forest plot of changes in kicking distance performance in athletes participating in plyometric training compared to controls.** Values shown are effect sizes (Hedges's g) with 95% confidence intervals (CI). The size of the plotted squares reflects the statistical weight of the study.

| Study name | Statistics for each study | | | | | | | Hedges's g and 95% CI |
|---|---|---|---|---|---|---|---|---|
| | Hedges's g | Standard error | Variance | Lower limit | Upper limit | Z-Value | p-Value | |
| Guadie [46] | 1.044 | 0.439 | 0.193 | 0.183 | 1.906 | 2.377 | 0.017 | |
| Sharma and Multani [77] | 1.284 | 0.342 | 0.117 | 0.614 | 1.953 | 3.759 | 0.000 | |
| Caliskan and Arikan [99] | 0.187 | 0.388 | 0.151 | -0.573 | 0.948 | 0.483 | 0.629 | |
| | 0.847 | 0.344 | 0.118 | 0.173 | 1.521 | 2.462 | 0.014 | |

**Fig 5. Forest plot of changes in speed dribbling performance in athletes participating in plyometric training compared to controls.** Values shown are effect sizes (Hedges's g) with 95% confidence intervals (CI). The size of the plotted squares reflects the statistical weight of the study.

(<18 years of age, ES = 0.29; ≥ 18 years of age, ES = 0.52; $p$ = 0.449), and throwing velocity (<18 years of age, ES = 0.69; ≥ 18 years of age, ES = 0.86; $p$ = 0.587).

Regarding training-related moderator variables, significantly greater improvements in throwing velocity were observed after a period of PT lasting > 7 weeks compared to those lasting ≤ 7 weeks (> 7 weeks, ES = 1.05; ≤ 7 weeks, ES = 0.29; $p$ = 0.011). However, no significant improvements were observed following PT when athletes performed > 7 weeks, as opposed to when they performed ≤ 7 weeks, for kicking velocity (> 7 weeks, ES = 0.50; ≤ 7 weeks, ES = 0.29; $p$ = 0.501). Moreover, no significant improvements were noted following PT when athletes performed >14 total PT sessions, as opposed to when they performed ≤ 14 total PT sessions, for kicking velocity (> 14 total PT sessions, ES = 0.51; ≤ 14 total PT sessions, ES = 0.30; $p$ = 0.506). Furthermore, no significant subgroup difference was found when comparing PT interventions with 2 sessions per week to those with 3 sessions per week for throwing velocity (2 sessions, ES = 0.99; 3 sessions, ES = 0.50; $p$ = 0.106). In addition, no significant subgroup differences were identified between sessions lasting > 30 minutes and those lasting ≤ 30 minutes (> 30 minutes, ES = 0.26; ≤ 30 minutes, ES = 0.40; $p$ = 0.690), and throwing velocity (> 30 minutes, ES = 0.53; ≤ 30 minutes, ES = 1.11; $p$ = 0.061).

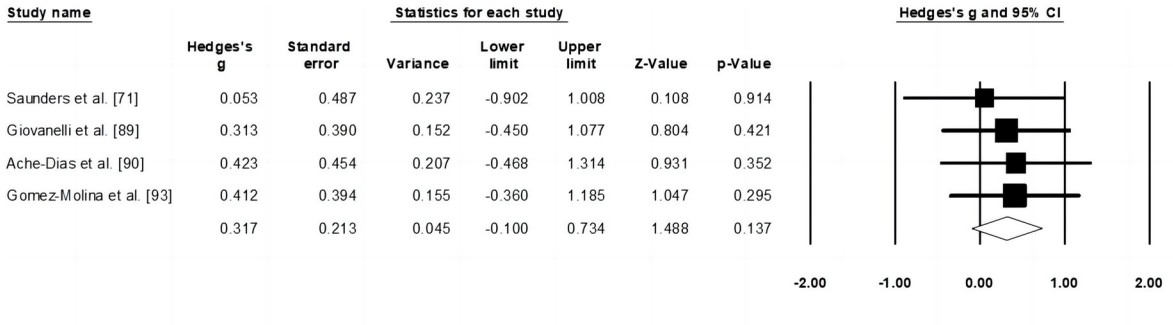

| Study name | Statistics for each study | | | | | | | Hedges's g and 95% CI |
|---|---|---|---|---|---|---|---|---|
| | Hedges's g | Standard error | Variance | Lower limit | Upper limit | Z-Value | p-Value | |
| Saunders et al. [71] | 0.053 | 0.487 | 0.237 | -0.902 | 1.008 | 0.108 | 0.914 | |
| Giovanelli et al. [89] | 0.313 | 0.390 | 0.152 | -0.450 | 1.077 | 0.804 | 0.421 | |
| Ache-Dias et al. [90] | 0.423 | 0.454 | 0.207 | -0.468 | 1.314 | 0.931 | 0.352 | |
| Gomez-Molina et al. [93] | 0.412 | 0.394 | 0.155 | -0.360 | 1.185 | 1.047 | 0.295 | |
| | 0.317 | 0.213 | 0.045 | -0.100 | 0.734 | 1.488 | 0.137 | |

**Fig 6. Forest plot of changes in stride rate performance in athletes participating in plyometric training compared to controls.** Values shown are effect sizes (Hedges's g) with 95% confidence intervals (CI). The size of the plotted squares reflects the statistical weight of the study.

## Adverse effects

No study in the analysis revealed adverse effects such as discomfort, pain, fatigue, injury, harm, or other health problems linked to the PT interventions used.

## Discussion

This systematic review and meta-analysis aimed to investigate the impact of PT on technical skill performance among athletes. The final analysis included 30 studies that were eligible for inclusion based on the selection criteria. The main finding of this study indicates that PT interventions induced small-to-moderate improvements (ES = 0.67 to 1.47) in kicking velocity and distance, throwing velocity, and speed dribbling, while a non-significant small improvement in stride rate was noted. Mostly, the level of heterogeneity in the above-mentioned results was low-to-moderate ($I^2$ = 0.0–57.6%). This finding aligns with previous reviews [101–103] that supported the efficacy of PT in enhancing athletic performance in athletes.

### The effect of PT on throwing velocity

Numerous sports and work-related tasks necessitate forceful movements involving the arms and hands [104]. In the current review, nine studies were included which examined the impact of PT on ball-throwing velocity in various sports. The participants of these studies included baseball players [72, 76], water polo players [82, 86, 87], handball players [81, 97, 98], and tennis players [19]. Our meta-analysis revealed moderate (ES = 0.78) PT-related improvements in measures of throwing velocity compared to active controls. Indeed, most (7 out of 9) of the included studies in the meta-analysis incorporated upper-body PT drills. These results concur with Singla et al.'s findings [105], which summarized the effects of upper-body PT in improving ball-throwing distance and velocity in healthy individuals. Upper-body PT utilizes medicine ball exercises and plyometric push-ups to enhance players' throwing muscles [106, 107]. Previous research has indicated that medicine ball exercises can lead to significant improvements in the upper body strength and power variables, particularly in the shoulder muscles, as well as rotational strength of the trunk and hip muscles in handball [108] and baseball players [109]. Internal and external rotator muscles, especially in the shoulders, are essential in ball speed [110]. Moreover, this exercise requires coordinating the agonist and antagonist muscles to maintain a rhythmic plyometric motion by incorporating a high-intensity concentric contraction right after an eccentric contraction [111]. Consequently, the nervous system is trained to respond faster [112]. Previously, overhead-throwing athletes were advised to follow the ballistic six exercise regimen [113, 114]. In a study by Carter et al. [72], when compared to the CG, baseball players that participated in an upper limb PT program employing six ballistic exercises increased throwing velocity. To simulate the actions, positions, and forces associated with the overhead throwing motion, the ballistic six-upper limb PT protocol includes a series of functional exercises carried out at high volumes [72]. In other words, to utilize the stretch reflex, plyometric exercises performed ballistic and explosive to reduce the amortization phase of the SSC. The results are reinforced by the findings of Grezios et al. [115], who showed that the SSC, which is the fundamental principle of PT, is the dominant type of muscle contraction used in overhead throwing. Interestingly, one study in this review reported that upper body PT could improve handball throwing speed, but no notable difference was detected between EG and CG. A similar duration of routine handball training also positively affects throwing speed [98]. Future research could explore the optimal combination of PT and handball training to maximize the benefits. Moreover, Chelly and colleagues [81] emphasized the importance of lower limb strength and power for throwing ability, suggesting that coaches should incorporate training programs for not only the upper body, but also the lower limbs. For

example, water polo players' throwing speed is influenced by multiple factors, including throwing technique, upper-body, lower-body, and trunk strength, and vertical jumping ability [116–118]. In agreement, two studies reviewed in this paper showed that either a combined upper and lower limb PT program or a lower limb PT program resulted in significant improvements in water polo players' overhead throwing velocity [86, 87]. The strong correlation between force and throwing velocity provides evidence that lower-body force can also influence the ability to generate throwing velocity by improving the capacity to push the body out of the water [119]. Additionally, in tennis, the serve can be compared to a throwing motion, and coaches often use throwing skills to train players to improve their serve [120]. Research suggests that exercises mimicking throwing actions with a SSC are more effective than isokinetic exercises for improving serve performance [121, 122]. Baiget et al. [123] suggest that upper-body plyometric exercises are effective for improving serve performance as they involve multiple body structures and can generate high force in short periods of time, which is essential for a fast serve. To achieve the best results, these exercises should be performed in a full range of motion, with high-velocity rotations, and involving multiple joints, particularly around the shoulder complex [123].

## The effect of PT on kicking velocity and distance

Several researchers have emphasized that kicking is critical in soccer [124–126]. Our meta-analysis noted that PT could help soccer players improve their technical skill performance in kicking velocity (ES = 0.37) [74, 75, 83–85, 91, 95, 100] and distance (ES = 0.44) [78–80, 92, 94, 96]. The effectiveness of a kick depends on various factors such as the maximum strength of the muscles involved [127, 128], the coordination between the nervous system and muscles, and the velocity at which the ankle moves both linearly and angularly in the kicking leg [129]. Markovic [130] stated in meta-analysis research that the benefits of plyometric exercises on vertical jump ability might transfer positively to sport-specific performances, such as kicking. The gains obtained in the kicking ability assessment might be attributed to PT-related neuromuscular adaptations of lower limb strength and power increases [23, 78, 125]. However, Campo et al. [74] and Sedano et al. [75] reported that six weeks of PT cannot improve soccer players' kicking speed. The PT intervention program likely resulted in a higher ball speed due to enhanced energy transfer from the closer to the farther segments. Consequently, athletes might need to allow sufficient time for their explosive strength gains to translate into kinematic factors, ultimately leading to a remarkable increase in kicking speed [131, 132]. This result might explain why no substantial gains in kicking speed occurred following six weeks of PT [74]. The results of these studies suggest that an effective PT program lasting over six weeks could potentially enhance the explosive strength of soccer players. More importantly, these enhancements can be transferred to soccer kicks in terms of ball velocity. Furthermore, these neuromuscular adaptations likely affected the kicking performance's biomechanical elements, such as the toe, ankle, knee, and hip's maximum linear velocities upon ball contact, leading to superior ball kicking velocities and even maximal kicking distances [125]. In addition, multiple studies have highlighted the importance of balance for kicking performance in soccer [133–135]. For instance, Cerrah et al. [136] found a correlation between the kicking velocity of the dominant leg and the dynamic balance ability of both legs. Moreover, a growing body of research has demonstrated that PT programs can enhance the balance ability of players [30, 39, 137].

## The effect of PT on speed dribbling

All players need to possess dribbling skills, but these abilities are particularly advantageous for attacking players who intend to penetrate a tight defense by running through congested areas

[138]. We observed a moderate enhancement in the speed of dribbling (ES = 0.85) following PT among skilled players of basketball [77], handball [46], and soccer [99]. This improvement could be attributed to enhanced intermuscular coordination, speed, and precision of movement after plyometric exercises [139]. Specifically, speed dribbling is a technique used in handball where the player runs as fast as possible while keeping the ball out in front for close control [140]. It requires a combination of speed, agility, coordination, and ball control [140]. On the other hand, speed dribbling in soccer primarily relies on agility, which is defined as the ability to change direction quickly. There appears to be a positive correlation between agility and dribbling performance [141]. Similarly, it appears that PT had a more specific impact on basketball compared to other sports. The nature of basketball, which involves horizontal and lateral running and fast and quick movements between opposing players for ball crossing, may have led to greater responses to PT and subsequent enhancements in the change of direction [29], which could potentially benefit speed dribbling [141]. Furthermore, Apostolidis and Emmanouil [142] found that handgrip strength was a predictor of speed dribbling in basketball players. Based on the information mentioned above, it is crucial to implement effective training programs to enhance players' physical fitness and optimize their game performance, specifically in skills such as speed dribbling [143]. PT has been widely studied by researchers as a means to improve athletes' physical fitness [15, 45, 51]. Nonetheless, the ideal training volume for enhancing speed dribbling performance remains undetermined and requires further investigation. Consequently, it can be concluded that an effective dribbling technique plays a crucial role in determining the outcome of a match [144]. However, it is surprising that there are very few studies examining the impact of PT on ball players' dribbling performance. Therefore, more research is needed to draw more reliable conclusions in this area.

## The effect of PT on stride rate

The product of stride length and stride frequency is commonly used to define running speed [145]. Schubert et al. [146] suggested that changes in stride rate (i.e., shorter strides) could affect impact peak, kinetics, and kinematics, and therefore, might be considered a mechanism for modifying injury risk and facilitating recovery in runners. However, our meta-analysis found no significant effect of PT on runners' stride rate performance [71, 89, 90, 93]. Similarly, a recent meta-analysis reported a non-significant impact (ES = 0.37) of jump training on stride rate in endurance runners [52]. The literature states that running speed improvement is usually accomplished by increasing stride length rather than stride frequency [147, 148]. Therefore, it is not unexpected that PT can enhance time trial performance without a significant improvement in stride rate performance. Furthermore, there is currently no evidence in the literature to suggest that neuromuscular adaptations resulting from plyometric exercises lead to changes in running biomechanics, such as stride rate [149]. In this sense, the lack of specificity in the PT program may have contributed to the non-significant effect on stride rate performance. Future studies should aim to investigate which types of plyometric jump drills (e.g., horizontal, vertical) or their combination with other training methods can effectively improve stride rate measures.

## Additional analysis

Our study conducted subgroup analyses to investigate whether factors such as the athlete's sex, age, and training variables had any impact on the effect of PT on their technical skill performance. However, our findings suggested that these factors did not have any significant effect on the enhancement of kicking velocity and distance, throwing velocity, speed dribbling, and stride rate performance after PT intervention, except for the subgroup analysis based on the length of the PT program. We found that longer PT interventions (over 7 weeks) resulted in

greater improvements in throwing velocity. The results of our study align with previous meta-analyses that reported no significant impact of sex, age, and training variables on the effectiveness of PT in enhancing physical fitness in athletes [16, 44]. Moreover, our findings support the notion that longer PT interventions may lead to greater improvements in throwing velocity among athletes, which is consistent with previous research showing that longer training interventions result in more significant physical fitness gains [20, 150]. Longer PT interventions may enable athletes to perform more volume of exercise drills, resulting in greater improvements in throwing speed. Indeed, previous literature has demonstrated a direct correlation between physical fitness and technical skill [42, 151]. In addition, our investigation into moderating factors on technical skill performance after PT may have been restricted by the preliminary nature of some of the studies. Hence, we cannot currently provide conclusive recommendations to athletes regarding the optimal training variables for PT to enhance their technical skills. Future research is necessary to determine the most effective PT strategies based on both athlete-specific and training-specific factors.

## Limitations

There are several notable limitations of this systematic review that should be taken into consideration. Firstly, although including a significant number of studies on a diverse range of sports, this review did not cover other power-strength related sports like badminton, ice hockey, and martial arts. Secondly, due to the limited number of studies, the meta-analysis of the effects of PT on outcomes such as handball passing accuracy [46], swimming block start [73], and gymnastics halt vault outcomes [88] was precluded. Moreover, this review only includes three studies that specifically focus on female players, which restricts our understanding of the general efficacy of PT in improving technical skill performance in athletes. Thirdly, we were unable to perform additional analyses on PT frequency, length, total sessions, and weekly session time in some cases due to the limited availability of at least one of the moderators (less than three papers). Therefore, we cannot provide definitive recommendations on the optimal training variables for improving athletes' technical skills. Fourthly, the use of a median split strategy to dichotomize continuous data (e.g., $> 7$ weeks vs. $\leq 7$ weeks) may result in residual confounding and reduced statistical power. Fifthly, the restriction of the publication search to studies written in English may have confined the findings' representation. Finally, according to the GRADE assessment, the level of certainty of the evidence for most of the study outcomes ranged from low to moderate, lowering confidence in the reported estimates.

## Conclusions

Our findings have shown that PT can be effective in enhancing technical skills measures in youth or adult athletes. Notably, PT has demonstrated significant improvements in kicking velocity and distance (i.e., soccer), throwing velocity (i.e., handball, baseball, water polo, tennis), and speed dribbling performance (i.e., soccer, handball, basketball). However, no significant effects on runners' stride rate performance were observed. Sub-group analyses suggest that longer PT ($> 7$ weeks) interventions appear to be more effective for improving throwing velocity. However, to fully determine the effectiveness of PT in improving sport-specific technical skill outcomes and ultimately enhancing competition performance, further high-quality research covering a wider range of sports is required.

## Practical application

This review's conclusions offer practical consequences for sports coaches, trainers, and players. PT may be recommended as a training strategy to improve technical skill performance,

especially for team sports athletes. Moreover, one significant advantage of this training is that it can be performed using inexpensive equipment such as medicine balls and jump boxes/hurdles. This feature makes it simple to incorporate into regular training regimens [10]. According to the results of the current review, there are no clear dose relationships evidence to recommend the optimal training variables to improve technical skill performance among athletes. As a general recommendation, trainers could consider exposing trainees to 1–3 sessions of 6–60 minutes each for a period of 4–16 weeks, as this time frame provides a suitable stimulus. Nevertheless, additional well-designed studies are necessary to identify the optimal dosages and analyze the interactions among training variables, with the goal of further enhancing the technical skills of the athletic population.

## Supporting information

**S1 Table. Detailed search strategy.**
(DOCX)

**S2 Table. Date used for meta-analysis.**
(DOCX)

**S3 Table. PRISMA 2020 checklist.**
(DOCX)

## Author Contributions

**Conceptualization:** Nuannuan Deng, Kim Geok Soh, Borhannudin Abdullah.

**Investigation:** Nuannuan Deng, Dandan Huang.

**Methodology:** Kim Geok Soh, Borhannudin Abdullah.

**Supervision:** Kim Geok Soh.

**Writing – original draft:** Nuannuan Deng.

**Writing – review & editing:** Kim Geok Soh, Borhannudin Abdullah, Dandan Huang, Wensheng Xiao, Huange Liu.

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
