## [Decision Letter · Decision Letter 0]

28 Apr 2023

PONE-D-23-09171Effects of Plyometric Training on Skill Performance among Athletes: A Systematic Review of Randomized Controlled TrialsPLOS ONE

Dear Dr. Deng,

Thank you for submitting your manuscript to PLOS ONE. After careful consideration, we feel that it has merit but does not fully meet PLOS ONE’s publication criteria as it currently stands. Therefore, we invite you to submit a revised version of the manuscript that addresses the points raised during the review process.

We look forward to receiving your revised manuscript.

Kind regards,

Bojan Masanovic, Ph.D.

Academic Editor

PLOS ONE

Journal Requirements:

Reviewers' comments:

Reviewer's Responses to Questions

**Comments to the Author**

1. Is the manuscript technically sound, and do the data support the conclusions?

Reviewer #1: Yes

Reviewer #2: No

Reviewer #3: Yes

2. Has the statistical analysis been performed appropriately and rigorously? 

Reviewer #1: Yes

Reviewer #2: No

Reviewer #3: Yes

3. Have the authors made all data underlying the findings in their manuscript fully available?

Reviewer #1: Yes

Reviewer #2: No

Reviewer #3: Yes

4. Is the manuscript presented in an intelligible fashion and written in standard English?

Reviewer #1: Yes

Reviewer #2: Yes

Reviewer #3: No

5. Review Comments to the Author

Reviewer #1: The study is methodologically very well set up and done. The PRISMA protocol was adequately used. A large number of studies were processed, the inclusion and exclusion criteria were well set. However, there are some things that need to be improved:

1. Do not use keywords that are in the title

2. Line 49 - the full name must be written before the abbreviation PT, because it is mentioned here for the first time in the text

3. In Table 2, why is there a different way of citing studies in the first column

4. In the Results section, there are a lot of statistical significance values written in parentheses, which are unnecessary and burden the work.

5. Line 286 - this sentence is already in the previous one, don't repeat

Reviewer #2: I read with interest the manuscript titled “Effects of Plyometric Training on Skill Performance among Athletes: A Systematic Review of Randomized Controlled Trials”. The study aimed to compile and synthesize the existing studies on the effects of plyometric training on healthy athletes’ skill performance. However, several minor and major critical issues preclude this reviewer from providing a favorable assessment.

Introduction

The introduction does not provide a clear conceptual-operational definition for “skill performance”. This is a critical issue for the rest of the study protocol. Similarly, a definition for “plyometric” would be advisable, and how “plyometric” jump-related and upper-body training exercises are different from other plyometric-related exercises (e.g., sprints).

Protocol and registration section

Major critical comment: the authors indicated “The PRISMA statement was followed in the reporting of this systematic review”. However, the authors did not comply with several PRISMA items. For example: i) the sixty-one publications excluded after a full-text examination were not reported, with figure 1 providing only generic exclusion reasons; ii) the authors stated that “The heterogeneity across measurement and training programs made it impossible to conduct a meta-analysis”. However, an actual heterogeneity analysis of the evidence (e.g., application of I2 statistics) was not provided. Several studies reported the same outcome, probably allowing a meta-analysis; and several statistical techniques (e.g., moderator analyses) may aid reduce heterogeneity. Further, in the register of the protocol (INPLASY) the authors did not include a meta-analysis; iii) A GRADE analysis was not included; iv) only articles in English language were included; v) the protocol was registered after the actual date of the literature search. Moreover, the protocol register provides only generic information (i.e., the register is an idea rather than a project).

Eligibility Criteria

The authors did not provide a detailed clarification for the Outcomes being analyzed. As for my comment related to the introduction section, this is a critical issue.

Search Strategy and Selection Process

I tried to replicate the results obtained by the authors, by using the search strategy provided in Table S1, but I was unable to, particularly for Scopus database. In addition, for the database WOS, is uncertain the collection (e.g., Core?) used for the search. Moreover, is not clear how the authors used the database Google Scholar to search for potential studies.

Further, a major critical issue in this systematic review are the keywords selected by the authors. These probably reduced the chances to find currently available studies includable for this systematic search. For example, the author’s included studies where the “skill” outcome was running (endurance) performance (time). Just taking this outcome in consideration, a significant number of studies were left out from the systematic review (please see this study: doi: 10.1080/02640414.2021.1916261).

Data extraction

Not clear why the authors did not consider the description of data extraction for outcomes in this section.

Results (and following section)

Considering my previous observations, I judge to be unsuitable the assessment of the results section (and following sections) due to potentially biased findings.

Reviewer #3: the manuscript is written very well, and methodologically very precise. The authors very thoroughly explained the methodology they used, as well as the results they reached. what needs to be corrected is the English language, which is not appropriate in some situations.

after that, the paper can be accepted for publication.

6. PLOS authors have the option to publish the peer review history of their article (what does this mean?). If published, this will include your full peer review and any attached files.

Reviewer #1: **Yes: **Jovan Gardasevic

Reviewer #2: No

Reviewer #3: No

---

## [Author Response · Author response to Decision Letter 0]

15 May 2023

Response to Reviewers

Dear reviewer,

We feel great thanks for your professional review work on our article. According to your nice suggestions, we have made extensive corrections to our previous manuscript, the reviewer specific concerns have been numbered. Due to the extensive revisions, we did not use the “track changes” function, and all the changes were marked in red in the resubmitted manuscript.

Reviewer #1:

1.Do not use keywords that are in the title

Response 1:

Thank you for your suggestion. We have changed our keywords, (i.e., plyometric exercise; sports; skill; athletic performance; stretch-shortening cycle)

2.Line 49 - the full name must be written before the abbreviation PT, because it is mentioned here for the first time in the text.

Response 2: 

Thank you for careful reading. We have revised this point, line 78

3.In Table 2, why is there a different way of citing studies in the first column

Response 3: 

Thank you for pointing this out. We have revised it in the resubmitted manuscript. Please see Table 3

4.In the Results section, there are a lot of statistical significance values written in parentheses, which are unnecessary and burden the work.

Response 4: 

Thank you for your valuable suggestion. Based on comments from other reviews, we have revised this part and added meta-analysis in there.

5.Line 286 - this sentence is already in the previous one, don't repeat

Response 5: 

Thank you for careful reading, Based on other reviews comments, we have revised this part

Reviewer #2: 

1.Introduction

The introduction does not provide a clear conceptual-operational definition for “skill performance”. This is a critical issue for the rest of the study protocol. Similarly, a definition for “plyometric” would be advisable, and how “plyometric” jump-related and upper-body training exercises are different from other plyometric-related exercises (e.g., sprints).

Response 1: 

Thank you for your careful review and valuable suggestion. Based on your suggestion, our research team had a discussion, we have revised the title of our review to include the term "technical skill performance," (Effects of Plyometric Training on Technical Skill Performance among Athletes: A Systematic Review and Meta-analysis), which more accurately reflects the focus of our work. Additionally, we have included a more comprehensive definition of technical skill and plyometric training in the introduction section to provide a clearer understanding of our research. Line 63-66, line 79-81, line 83-88.

Plyometric training is a type of strength training that mainly consists of various jumping, hopping, skipping, and throwing exercises. These exercises are inherent in most sports movements, such as high jumping, pitching, or kicking. However, the plyometric exercises used in a training program should match the individual needs of the athlete in relation to the characteristics of the sporting activity that they are involved with. That is, to optimize transference to sport plyometric exercises should reflect the type of activity implicit in that sport, that is, the principle of specificity. Such as, sprint performance gains will be optimized by the use of training programs that incorporates greater horizontal acceleration (i.e., sprint-specific plyometric exercises, jumps with horizontal displacement).

We have added these information in introduction section, line 83- 101.

2.Protocol and registration section

Major critical comment: the authors indicated “The PRISMA statement was followed in the reporting of this systematic review”. However, the authors did not comply with several PRISMA items. For example: 

Response 2: 

We sincerely appreciate your comments, we have revised the PRISMA checklist (S3 table). and the content following the updated PRISMA statement.

2.1The sixty-one publications excluded after a full-text examination were not reported, with figure 1 providing only generic exclusion reasons; 

Response 2.1: 

We sincerely appreciate your comments. We re-searched and carefully screened the literature. We have revised the PRISMA table (Fig 1), and included detailed reasons for excluding studies. Please see Figure 1

2.2The authors stated that “The heterogeneity across measurement and training programs made it impossible to conduct a meta-analysis”. However, an actual heterogeneity analysis of the evidence (e.g., application of I2 statistics) was not provided. Several studies reported the same outcome, probably allowing a meta-analysis; and several statistical techniques (e.g., moderator analyses) may aid reduce heterogeneity.

Response 2.2: 

Thank you for your careful review and valuable suggestion. We have conducted meta-analysis in our resubmitted paper. And here we did not list the changes but marked in red in the revised manuscript.

2.3 Further, in the register of the protocol (INPLASY) the authors did not include a meta-analysis; 

Response 2.3: 

We sincerely appreciate your suggestion. We have updated the review protocol with a complete description of the research process and added a meta-analysis. (https://inplasy.com/?s=INPLASY202320052)

2.4 A GRADE analysis was not included; 

Response 2.4:

Thank you for your comment. We have added the GRADE approach to assess and summarize the certainty of evidence in included papers. Line 249- 267, line 338 - 342, the GRADE assessment can be found in table 2.

2.5 Only articles in English language were included; 

Response 2.5:

Thank you for carefully reviewing, considering the potential difficulty of translating articles into different languages and the fact that 99.6% of plyometric jump training literature is published in English (Ramírez-Campillo et al., 2018), Moreover, several plyometric training reviews also only selected papers written in English, thus, only articles written in English were considered for our review. We have added this statement to the eligibility criteria section. Line 208 - line 210.

 (Ref: Ramirez-Campillo, R., Alvarez, C., Garcia-Hermoso, A., Ramirez-Velez, R., Gentil, P., Asadi, A., et al. (2018). Methodological characteristics and future directions for plyometric jump training research: a scoping review. Sports Med. 48, 1059–1081. doi: 10.1007/s40279-018-0870-z)

2.6 The protocol was registered after the actual date of the literature search. Moreover, the protocol register provides only generic information (i.e., the register is an idea rather than a project).

Response 2.3:

Thank you for your careful reading and valuable suggestion, we have updated the review’s protocol with a full description of the methods on the INPLASY website (https://inplasy.com/?s=INPLASY202320052).

3.Eligibility Criteria

The authors did not provide a detailed clarification for the Outcomes being analyzed. As for my comment related to the introduction section, this is a critical issue.

Response 2.4: 

We sincerely appreciate your suggestion. We have added a detailed definition for the outcomes being analyzed (i.e., technical skill performance) in eligibility criteria section, line173-179 

4.Search Strategy and Selection Process

4.1I tried to replicate the results obtained by the authors, by using the search strategy provided in Table S1, but I was unable to, particularly for Scopus database. In addition, for the database WOS, is uncertain the collection (e.g., Core?) used for the search. Moreover, is not clear how the authors used the database Google Scholar to search for potential studies.

Response 4.1: 

Thank you for your comment. We redeveloped the search strategy and double-checked the repeatability of the search strategy in all databases, it can be found in S1 Table. We searched the WOS core collection, Furthermore, relevant supplementary material was searched for through manual searches on Google Scholar, including article citations and free-text searches. we have revised the statement in resubmitted paper, line 198-200, line 189

4.2Further, a major critical issue in this systematic review are the keywords selected by the authors. These probably reduced the chances to find currently available studies includable for this systematic search. For example, the author’s included studies where the “skill” outcome was running (endurance) performance (time). Just taking this outcome in consideration, a significant number of studies were left out from the systematic review (please see this study: doi: 10.1080/02640414.2021.1916261). Effects of jump training on physical fitness and athletic performance in endurance runners: A meta-analysis.

Response 4.2: 

Thanks so much for the advice and great help. We have revised our keywords and search strategy, line 193-195. 

Moreover, based on the definition of technical skill performance, studies that solely assessed time trial performance outcomes (e.g., running) rather than sport-specific technical skills (e.g., stride frequency) were not considered. Furthermore, we manually searched the reference lists of this meta-analysis and other relevant review studies to discover any publications that were not detected by the initial computerized search. Line 196-203, line 177-179.

5.Data extraction

Not clear why the authors did not consider the description of data extraction for outcomes in this section.

Response 5: 

Thank you for your comment. We have revised this point. Line 228-232.

6.Results (and following section)

Considering my previous observations, I judge to be unsuitable the assessment of the results section (and following sections) due to potentially biased findings.

Response 6: 

Thanks so much for your comments. Based on your input, we have made revisions to this section and included a meta-analysis. We have also made changes to the discussion section in light of the results., please see the result and discussion section.

Reviewer #3: What needs to be corrected is the English language, which is not appropriate in some situations.

Response:

Thank you for careful reading. We apologize for the language problems in the original manuscript. The language presentation was improved with assistance from a native English speaker with appropriate research background. And we also carefully checked the entire manuscript for typographic, grammatical and formatting errors.

---

## [Decision Letter · Decision Letter 1]

26 Jun 2023

Effects of Plyometric Training on Technical Skill Performance among Athletes: A Systematic Review and Meta-analysis

PONE-D-23-09171R1

Dear Dr. Deng,

We’re pleased to inform you that your manuscript has been judged scientifically suitable for publication and will be formally accepted for publication once it meets all outstanding technical requirements.

Kind regards,

Bojan Masanovic, Ph.D.

Academic Editor

PLOS ONE

Additional Editor Comments (optional):

Reviewers' comments:

Reviewer's Responses to Questions

**Comments to the Author**

1. If the authors have adequately addressed your comments raised in a previous round of review and you feel that this manuscript is now acceptable for publication, you may indicate that here to bypass the “Comments to the Author” section, enter your conflict of interest statement in the “Confidential to Editor” section, and submit your "Accept" recommendation.

Reviewer #1: All comments have been addressed

Reviewer #3: All comments have been addressed

2. Is the manuscript technically sound, and do the data support the conclusions?

Reviewer #1: Yes

Reviewer #3: Yes

3. Has the statistical analysis been performed appropriately and rigorously? 

Reviewer #1: Yes

Reviewer #3: Yes

4. Have the authors made all data underlying the findings in their manuscript fully available?

Reviewer #1: Yes

Reviewer #3: Yes

5. Is the manuscript presented in an intelligible fashion and written in standard English?

Reviewer #1: Yes

Reviewer #3: Yes

6. Review Comments to the Author

Reviewer #1: The authors corrected everything that was required of them in the revised manuscript. Only one thing is missing. To my comment about keywords I can't see keywords in the manuscript now?

Reviewer #3: (No Response)

7. PLOS authors have the option to publish the peer review history of their article (what does this mean?). If published, this will include your full peer review and any attached files.

Reviewer #1: **Yes: **Jovan Gardasevic, University of Montenegro

Reviewer #3: **Yes: **Milovan Ljubojevic

---

## [Editor Report · Acceptance letter]

7 Jul 2023

PONE-D-23-09171R1 

Effects of Plyometric Training on Technical Skill Performance among Athletes: A Systematic Review and Meta-analysis 

Dear Dr. Deng:

I'm pleased to inform you that your manuscript has been deemed suitable for publication in PLOS ONE. Congratulations! Your manuscript is now with our production department. 

Kind regards, 

on behalf of

Dr. Bojan Masanovic 

Academic Editor

PLOS ONE